# Revealing the 3D Cosmic Web through Gravitationally Constrained Neural Fields

**Brandon Zhao**[1*]  **Aviad Levis**[2,3]  **Liam Connor**[4]  **Pratul P. Srinivasan**[5]  **Katherine L. Bouman**[1,6]

[1]Department of Computing and Mathematical Sciences, California Institute of Technology
[2]Department of Computer Science, University of Toronto
[3]David A. Dunlap Department of Astronomy & Astrophysics, University of Toronto
[4]Center for Astrophysics, Harvard & Smithsonian
[5]Google DeepMind
[6]Departments of Astronomy and Electrical Engineering, California Institute of Technology
[*]`byzhao@caltech.edu`

## Abstract

Weak gravitational lensing is the slight distortion of galaxy shapes caused primarily by the gravitational effects of dark matter in the universe. In our work, we seek to invert the weak lensing signal from 2D telescope images to reconstruct a 3D map of the universe's dark matter field. While inversion typically yields a 2D projection of the dark matter field, accurate 3D maps of the dark matter distribution are essential for localizing structures of interest and testing theories of our universe. However, 3D inversion poses significant challenges. First, unlike standard 3D reconstruction that relies on multiple viewpoints, in this case, images are only observed from a single viewpoint. This challenge can be partially addressed by observing how galaxy emitters throughout the volume are lensed. However, this leads to the second challenge: the shapes and exact locations of unlensed galaxies are unknown, and can only be estimated with a very large degree of uncertainty. This introduces an overwhelming amount of noise which nearly drowns out the lensing signal completely. Previous approaches tackle this by imposing strong assumptions about the structures in the volume. We instead propose a methodology using a gravitationally-constrained neural field to flexibly model the continuous matter distribution. We take an analysis-by-synthesis approach, optimizing the weights of the neural network through a fully differentiable physical forward model to reproduce the lensing signal present in image measurements. We showcase our method on simulations, including realistic simulated measurements of dark matter distributions that mimic data from upcoming telescope surveys. Our results show that our method can not only outperform previous methods, but importantly is also able to recover potentially surprising dark matter structures.

## 1 Introduction

According to Einstein's theory of general relativity, massive objects in our universe bend their surrounding spacetime, causing nearby light to travel on curved trajectories. As a result, images of faraway galaxies as observed from Earth are warped due to the continuous deflection of their light by intervening matter. This effect, called gravitational lensing, is a powerful probe of the distribution of the massive structures underlying our universe, referred to colloquially as the *cosmic web*. Although the mass of visible matter can be directly measured by spectroscopic methods (Tinsley (1972), Roberts (1962)), dark matter does not appear to interact with electromagnetic radiation and thus cannot be observed directly. Nonetheless, gravitational lensing is an effect caused by the total matter distribution in the universe, both dark and luminous. Because dark matter constitutes around 85% of total matter by recent estimates (Aghanim et al. (2020)), gravitational lensing serves as an effective tool for estimating its distribution. In this work, we focus on the weak lensing regime, where light deflections are too small to interpret for a single lensed galaxy. However, by observing how weak lensing subtly shears the images of galaxies in dense fields, we aim to reconstruct a map of the underlying matter.

Traditional weak lensing analysis is performed in 2D. Measured galaxy shears are interpolated to form a single 2D *shear map* which is used to recover an estimate of the 2D projected matter density

over the sky. Many methods for 2D density reconstruction have been proposed in the past (Kaiser & Squires (1993), Lanusse et al. (2016)). However, analysis from these maps is limited; 2D mapping techniques do not provide any information about the mass of structures or their distance from Earth.

We seek to tackle the problem of 3D mass mapping; that is, inverting the weak lensing signal from 2D telescope images to obtain a 3D reconstruction of the dark matter field. Detection and localization of peaks in the 3D structure of dark matter is important for the study of its fundamental properties, many of which are currently unknown. For example, an open question in physics is whether or not dark matter emits trace amounts of gamma ray radiation (Bringmann & Weniger (2012)); however, it is difficult to discriminate between potential weak dark matter emission and the overwhelming background of gamma rays without attenuating to regions with peaks in dark matter density. In addition, characterizing the exact nature of of the primordial matter field at times close to the Big Bang requires studying the 3D structure of the current universe further than just statistics from the 2D field (Bartolo et al. (2004)). Thus, studying the 3D structure of dark matter offers exciting avenues for answering fundamental questions about our universe.

Recovering the 3D matter density field from projected cosmic shear measurements poses several challenges. First, since galaxies are observed from a single viewing angle, reconstructing the 3D distribution from shear data is inherently ill-posed, meaning many different dark matter configurations could produce the same observational data. While observing galaxy emitters throughout the volume helps to alleviate this issue, it introduces an even bigger challenge: the true unlensed shapes of visible sources are largely unknown. Because weak lensing describes the change between the observed shape and this "intrinsic" shape, this uncertainty introduces a nearly overwhelming amount of noise. For example, for a single galaxy, the noise from a galaxy's intrinsic shape can be on average 10 to 100 times the magnitude of the lensing shear signal.

We propose an approach that leverages the physics of gravitational lensing to recover a continuous 3D matter field represented using a neural field. In particular, we represent the spatially varying matter density field with a coordinate-based neural network that we optimize through a fully differentiable lensing forward model to reproduce a given set of observed shear measurements. We show on a simplified simulation of dark matter that our method not only accurately recovers 3D matter structures, but also surpasses the baseline in localizing these structures at their correct distances from Earth, referred to as *redshift*. Crucially, we also show that our method can accurately reconstruct non-Gaussian features not present in the simplified simulation where the baseline method struggles, which will prove essential for constraining fundamental theories of structure formation in the universe. We believe that our work serves as a flexible and powerful approach that shows promise for extension to upcoming weak lensing telescope surveys.

## 2 RELATED WORK

### 2.1 DARK MATTER MASS MAPPING

Dark matter mass mapping is the inverse problem of reconstructing the distribution of dark matter in the universe from observations of its gravitational lensing effects on images of galaxies. In weak lensing, which is the focus of this paper, the effects of lensing are very small and can be understood as small rotations or changes in the axis ratios of approximately elliptical galaxy shapes. These two effects are described by the complex components of the shear $\gamma$, a 2D field describing the spatial distribution of elliptical shape changes in galaxy images across the sky.

It was discovered by Kaiser & Squires (1993) that a dense shear field can be analytically inverted to obtain a 2D projected mass map of the sky. However, in practice we are limited in our ability to measure the full shear field by the visibility and density of galaxies in the sky as well as uncertainty in galaxies' unlensed shapes. Thus the problem of obtaining the 2D projected matter density is in itself difficult. Many methods for solving this 2D inversion have been developed: Kaiser & Squires (1993) propose a smoothing of the observed shear field before directly applying the analytical inversion. Regularized approaches such as a Gaussian prior (Horowitz et al. (2019)) and sparsity (Lanusse et al. (2016)) have also been used. However, all these methods are fundamentally limited in that 2D mass maps do not give information on the exact mass or redshift of detected structures. Only by measuring galaxy redshifts to reason in 3D can we estimate these quantities.

In the problem of 3D mass mapping we seek to recover not only the projected mass density but its full 3D distribution. This amounts to solving an ill-posed single-view tomography along the line of sight. Thus, some form of regularization is necessary for reconstruction. Previous work in 3D reconstruction from weak lensing has focused on methods that incorporate very strong priors on the spatial structure of the universe. Simon et al. (2009) proposed a Wiener filter method that combines a Gaussian prior with an inverse variance filter to regularize a reconstruction. VanderPlas et al. (2011) use an inverse variance filter as well, but regularize by truncating the singular value decomposition of their reconstruction. While both methods introduce heavy smoothing of their reconstructions along the line of sight, Leonard et al. (2014) proposed a method to achieve higher resolution in the line of sight direction by using a sparsity prior. However, this method induces a strong constraint, i.e. that the universe can be represented as a sparse sum of dark matter "halos", and can be susceptible to false peak detections along the line of sight. In addition, the optimal amount of sparsity is not clear, and different values can result in significantly different reconstructions.

Out of the aforementioned 3D reconstruction methods, the Wiener filtering method that employs a Gaussian prior based on the power spectrum of the dark matter field (Simon et al. (2009)) has been the most widely used in real data (Oguri et al. (2018), Simon et al. (2012)). Assuming that the underlying dark matter field is Gaussian, this method is favorable in that it is provably the optimal reconstruction method. However, there is currently a plethora of evidence pointing towards the non-Gaussian nature of the universe's density field, especially at low redshift regions close to Earth (Bartolo et al. (2004)). We show in this work that our method can reconstruct non-Gaussian structures more accurately, which will become critical for studying non-Gaussian features of the universe.

## 2.2 COORDINATE-BASED NEURAL FIELDS

Coordinate-based Neural Fields (Xie et al. (2022)), also commonly referred to as coordinate-based neural representations, have gained popularity in computer vision and graphics across a wide range of inverse problems. These models parameterize a continuous three-dimensional vector or scalar field with the weights of a multi-layer perceptron (MLP) that takes a spatial coordinate as input and outputs the value of the field at that coordinate. Typical approaches using neural fields formulate inverse problems as an optimization, directly using an underlying physical forward model to recover a single solution field that matches a set of observed measurements.

For ill-posed inverse problems where there are potentially many solution fields which fit a set of observed measurements, neural fields have been shown to provide a good implicit prior. In particular, MLPs favor learning lower spatial frequencies when optimized with gradient descent (Rahaman et al. (2019)), an effect known as spectral bias. Tancik et al. (2020) conducted a theoretical analysis which showed that by prepending a positional encoding layer one can modify the network's bandwidth in a tunable fashion. Thus, a positionally encoded MLP implicitly imposes a smoothness prior on the final reconstruction as it will favor representing certain spatial frequencies of a signal. Neural fields with positional encoding have been shown to provide impressive results on a wide range of ill-posed inverse problems, such as medical imaging (Shen et al. (2022)), cryo-electron microscopy (Zhong et al. (2021)), refractive field estimation (Zhao et al. (2024)), and black hole tomography (Levis et al. (2022), Levis et al. (2024)). Recently, Zhao et al. (2024) presented an approach to 3D dark matter mass mapping that optimizes a neural field model to fit to simulated weak lensing image measurements and showed how it led to superior performance over a traditional voxel grid. While this method is relevant to a larger range of gravitational lensing regimes, as it doesn't require lensed galaxies to be modeled as ellipses, it struggles when there is noise on the intrinsic shape of galaxies. In contrast, our work focuses on the noise-dominated weak lensing regime and introduces a method that takes advantage of fitting directly to the estimated shear of elliptical galaxies.

# 3 METHODS

## 3.1 WEAK LENSING MEASUREMENTS

As light travels towards us from faraway galaxies, its path is continuously perturbed by the dark matter it encounters along the line of sight. In weak lensing, the degree of this perturbation is relatively minor, and results in small changes in the observed shapes of galaxies. To quantify these small changes, approximately elliptical galaxy shapes are described using a complex ellipticity pa-

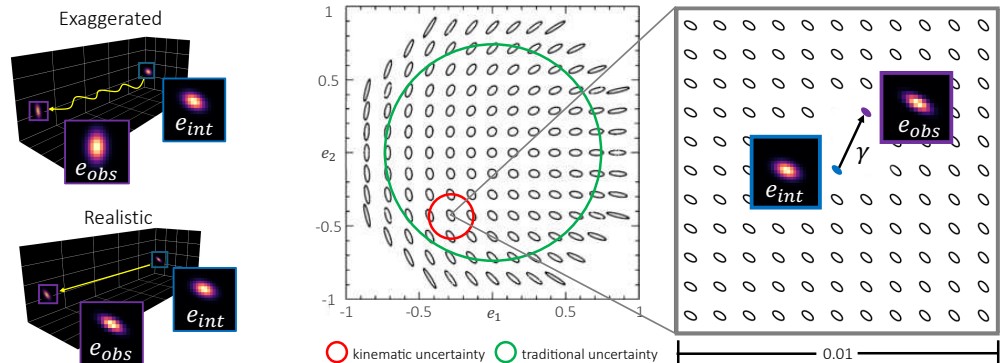

Figure 1: Weak Lensing Measurements. (Left) As light travels through the universe, it is lensed by intervening matter structures, causing slight changes between the unlensed intrinsic shape $e_{\text{int}}$ and the observed shape $e_{\text{obs}}$ of galaxies. Measurements of these faint shape changes is a useful probe of the 3D structure of dark matter. (Right) To quantify the effects of cosmic shearing, it is useful to consider galaxy shapes in the complex elliptical domain, where the components of the complex ellipticity describe the axis ratio and orientation angle of a given ellipse (Figure adapted from Schneider et al. (2006)). In kinematic weak lensing, the detectable cosmic shear signal is outweighed by our uncertainty in a galaxy's intrinsic shape by more than an order of magnitude. In traditional weak lensing the uncertainty is more than two orders of magnitude greater than the shear signal. The lensing effects of intervening matter can be described by a shear $\gamma$, which is approximately additive in the ellipticity domain. In practice, we combine many shear measurements from a dense field of galaxies to obtain a coherent signal from the underlying matter distribution.

rameterization $e$. Given an approximately elliptical galaxy with axis ratio $r$ and orientation angle $\phi$, we can define the magnitude and phase of its ellipticity $e$ as in Schneider et al. (2006):

$$|e| = \frac{1 - r}{1 + r} \qquad \angle e = 2\phi. \qquad (1)$$

Weak lensing measurements relate a galaxy's observed ellipticity $e_{\text{obs}}$ to its intrinsic ellipticity $e_{\text{int}}$, which describes what its observed shape would be in the absence of lensing. In the regime of weak lensing, the lensing effects of intervening matter can be approximately described as a shear $\gamma$, which is additive in the ellipticity domain (Schneider et al. (2006)):

$$e_{\text{obs}} - e_{\text{int}} = \gamma. \qquad (2)$$

An overview of shear measurements and the complex ellipticity can be found in Fig. 1. In practice, although the observed shape $e_{\text{obs}}$ can be measured directly from an image, a galaxy's intrinsic shape $e_{\text{int}}$ is much more difficult to estimate. In traditional weak lensing surveys, $e_{\text{int}}$ is completely unknown (Schneider et al. (2006)). More recently a technique called kinematic weak lensing (Huff et al. (2013)) has been proposed that leads to an estimate of $e_{\text{int}}$ using spectral data. Nonetheless, even with kinematic estimates of the intrinsic shape, the noise on $\gamma$ will still typically be more than 10 times larger than the signal itself; this uncertainty is called the shape noise. To mitigate the effects of the shape noise, typical weak lensing surveys combine measurements from many galaxies into a galaxy catalog containing the sky position, redshift (distance), and observed and intrinsic shape estimates of millions of galaxies, which constitute the measurements for the 3D mass mapping problem.

## 3.2 FORWARD MODEL

The shear observed in a galaxy image arises from the cumulative lensing occurring along the line of sight between us and the galaxy. Because a uniform sheet of mass does not induce any gravitational shearing effect, the shear is a measurement of the overdensity $\delta = (\rho - \bar{\rho})/\bar{\rho}$, where $\rho$ is the continuous matter density field and $\bar{\rho}$ is mean density at a given distance from Earth.

To compute the shear from the matter overdensity field $\delta$, we first introduce the convergence $\kappa$ (Schneider et al. (2006)), which for a galaxy with sky position $\boldsymbol{\theta}$ at distance $w$ is:

$$\kappa(\boldsymbol{\theta}, w) = \mathbf{Q}\delta = \frac{3H_0^2 \Omega_m}{2c^2} \int_0^w \mathrm{d}w' \frac{w'(w - w')}{w} \frac{\delta(w'\boldsymbol{\theta}, w')}{a(w')}, \qquad (3)$$

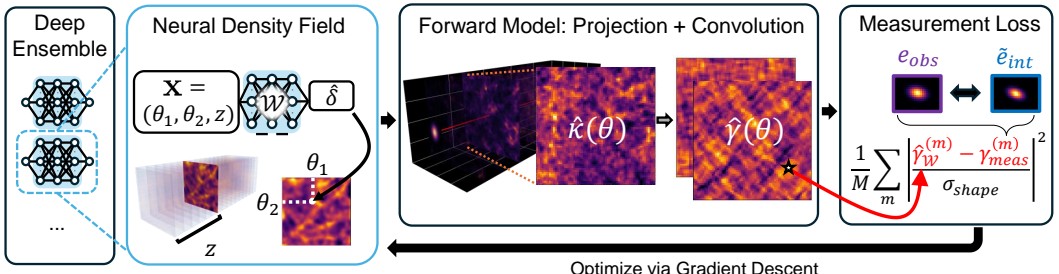

Figure 2: Proposed 3D Mass Mapping Pipeline. We model a 3D matter overdensity field $\hat{\delta}$ as a continuous function using a fully-connected neural network. We then differentiably compute cosmic shear measurements from a given galaxy catalog through this overdensity field with a physics-based forward model to produce a set of predicted shear measurements. Next, we solve for the weights of the neural network by minimizing a data loss between the model and observed shear measurements plus a physically-motivated power spectrum regularization loss. We obtain a final reconstruction by taking the median from a deep ensemble of 100 independently initialized neural fields.

where $H_0$ and $\Omega_m$ are assumed cosmological constants and $c$ is the speed of light. The scale parameter $a(w)$ represents the known expansion of the universe that occurs as light travels towards us, and $\delta$ is the matter overdensity. The 3D shear $\gamma$ is obtained by convolving the convergence $\kappa$ with a complex kernel $\mathcal{D}$ (Kaiser & Squires (1993)):

$$\gamma(\boldsymbol{\theta}, w) = \mathbf{P}\kappa = \frac{1}{\pi} \int_{\mathbb{C}}^{2} \mathrm{d}^2\boldsymbol{\theta}' \mathcal{D}(\boldsymbol{\theta} - \boldsymbol{\theta}')\kappa(\boldsymbol{\theta}', w), \ \mathcal{D}(\boldsymbol{\theta}) = \frac{-1}{(\boldsymbol{\theta}*)^2}. \tag{4}$$

Here, the angular position $\boldsymbol{\theta} = \theta_1 + i\theta_2$ is expressed in complex coordinates, and the asterisk $*$ represents complex conjugation.

Notably, the forward operators in Eqns. 3 and 4 are both linear with respect to the overdensity $\delta$, so obtaining the overdensity from the shear measurements is a linear inverse problem. Combining with Eqn. 2 gives us the full formulation with the effects of the additive shape noise $\varepsilon$:

$$e_{\text{obs}} = \mathbf{PQ}\delta + \varepsilon \qquad \varepsilon \sim \mathcal{N}(\tilde{e}_{\text{int}}, \sigma_{\text{shape}}), \tag{5}$$

where $\mathbf{Q}$ and $\mathbf{P}$ are the linear forward operators from Eqns. 3 and 4, respectively, and $\tilde{e}_{\text{int}}$ is the estimated intrinsic ellipticity. Note that even in traditional weak lensing, where no estimate of the intrinsic shape is available, we can set $\tilde{e}_{\text{int}} = 0$ and adjust $\sigma_{\text{shape}}$ to account for the full range of possible ellipses. In practice we calculate the forward model by discretizing the overdensity $\delta$ into a set of lens planes spaced throughout the volume. We then efficiently compute $\mathbf{Q}$ with summation, and the convolution $\mathbf{P}$ with the Fast Fourier Transform.

### 3.3 CONTINUOUS NEURAL REPRESENTATION

To solve the inverse problem of mass mapping, we take an analysis-by-synthesis approach. We model the continuous matter overdensity field $\delta$ with a neural field consisting of a coordinate-based Multi-Layer Perceptron (MLP), which is optimized to fit the observed measurements $e_{\text{obs}}$ through the forward model described in Sec. 3.2. The MLP, parameterized by weights $\mathcal{W}$, takes 3D spherical coordinates $\mathbf{x} = (\theta_1, \theta_2, w)$ as input and outputs a scalar matter density:

$$\hat{\delta}(\mathbf{x}) = \text{MLP}_{\mathcal{W}}(\mathcal{F}_L(\mathbf{x})), \ \mathcal{F}_L(\mathbf{x}) = [\sin(\mathbf{x}), \cos(\mathbf{x}), \ldots, \sin(2^{L-1}\mathbf{x}), \cos(2^{L-1}\mathbf{x})]^T, \tag{6}$$

where $\mathcal{F}_L(\mathbf{x})$ is a positional encoding layer mapping each input coordinate to a set of Fourier basis functions with increasing frequencies. The positional encoding layer has been shown to control the bandwidth of the neural network's interpolation kernel (Tancik et al. (2020)), tunable via the frequency parameter $L$. Higher levels of $L$ allow the MLP to express higher spatial frequencies, while lower levels of $L$ encourage learning lower spatial frequencies. In our experiments we use $L = 2$ for angular coordinates and $L = 5$ for the radial coordinate.

### 3.4 OPTIMIZATION

Because our network is a fully continuous estimate of the matter overdensity field $\delta$, we can simulate cosmic shear measurements that would have been observed had the galaxies in our catalog been

lensed by the density distribution represented by our neural field. To do so, we simply apply the forward operators $\mathbf{P}$ and $\mathbf{Q}$ detailed in Eqns. 4 and 3 to the neural overdensity estimate $\hat{\delta}$ to obtain the modeled shear $\hat{\gamma}$. We then optimize our network weights $\mathcal{W}$ to match the observed galaxy ellipticities $e_{\text{obs}}$ to the specified level of shape noise, minimizing a $\chi^2$ data loss.

We regularize our reconstruction with an additional loss term equal to the mean squared error between the lens plane power spectra of the ground truth ($P$) and the reconstruction ($\hat{P}$). Inclusion of this term helps us to prevent overfitting to the noise and produce physically realistic reconstructions. The theoretical form of the power spectrum $P$ can be analytically calculated and has been used to characterize the covariance of a Gaussian prior in previous work; however, we expect the ground truth field to contain non-Gaussian features in practice.

The full loss function is:

$$\mathcal{L}(\mathcal{W}) = \frac{1}{M} \sum_m \left| \frac{\hat{\gamma}_{\mathcal{W}}^{(m)} - \gamma^{(m)}}{\sigma_{\text{shape}}} \right|^2 + \frac{\lambda}{N} \sum_n \sum_\ell (\hat{P}_{\mathcal{W}}^{(n)}(|\ell|) - P^{(n)}(|\ell|))^2 \qquad (7)$$

where $m$ and $n$ index over $M$ galaxy catalog entries and $N$ lens planes, and $\ell$ is the frequency vector of a lensplane's 2D Fourier transform. A diagram of our model's optimization is shown in Fig. 2.

### 3.5 NEURAL DEEP ENSEMBLE

Because the forward operator $\mathbf{Q}$ is a weighted line-of-sight projection, solving the inverse problem of 3D mass mapping is akin to computing a tomography from a single line of sight. This, in combination with the large degree of noise in the lensing signal, makes the inversion ill-posed; there exist infinitely many mass fields that could produce the same set of shear observations. In particular, two neural networks with different sets of randomly initialized weights could potentially converge to different reconstructions, both of which fit a given observed galaxy catalog equally well. We find that constructing an ensemble estimator by taking the median values over many neural reconstructions gives robust reconstruction results and mitigates artifacts from individual samples[1].

In our experiments, we use a deep ensemble of 100 fully connected MLPs each with 4 layers, where each layer is 256 units wide with the ReLU activation function. The network output is passed through the sigmoid activation function followed by a final single node linear layer. We optimize the network weights by minimizing the loss in Eqn. 7 via gradient descent using the Adam optimizer (Kingma (2014)), with exponential learning rate decay from $1\mathrm{e}{-4}$ to $5\mathrm{e}{-6}$ over 100K iterations. All code is implemented in `JAX` and will be made publicly available.

## 4 EXPERIMENTS

### 4.1 COSMIC SHEAR SIMULATIONS

To simulate ground truth dark matter fields we used the low resolution Particle-Mesh N-body solver `JaxPM` (Initiative (2021)). The shear signal was simulated over a field of $5° \times 5°$ with a resolution of $75 \times 75$ pixels. For the kinematic weak lensing experiment in Sec. 4.2, we simulate shape noise in line with current estimates for the instrument capabilities of the Roman Space Telescope (Xu et al. (2023)), with a galaxy number density of $n_{\text{gal}} = 4$ arcmin$^{-1}$ and a shape noise level of $\sigma_{\text{shape}} = 0.035$. For the traditional weak lensing survey in Sec. 4.3, we assume no estimation has been done on the intrinsic shape of a denser field of galaxies, corresponding to a shape noise level of $\sigma_{\text{shape}} = 0.25$ and a galaxy number density of $n_{\text{gal}} = 30$ arcmin$^{-1}$. Finally, we assume a realistically distributed galaxy sample as in Leonard et al. (2014) of 360,000 galaxies for kinematic weak lensing and 2,700,000 galaxies in the traditional weak lensing experiment; more details are in the appendix.

Throughout this section, we will use the term *redshift*, which should be understood as interchangeable with distance in the radial direction; for reference, redshift $z = 0$ corresponds to Earth's location, while a redshift of $z = 2$ corresponds roughly to a distance of 15 billion light years away in the present day. For our experiments, we divide the lensing volume into 18 equally spaced lens planes from redshift $z = 0$ to 2. Although simulated galaxies are lensed throughout the whole volume, the SNR of the reconstruction decreases with distance from the sensor, as most of the lensing is done by

---

[1]Interestingly, although the individual members of the so-called deep ensemble do not correspond to samples from a Bayesian posterior, deep ensembles have been shown to yield predictive uncertainty estimates (Lakshminarayanan et al. (2017)).

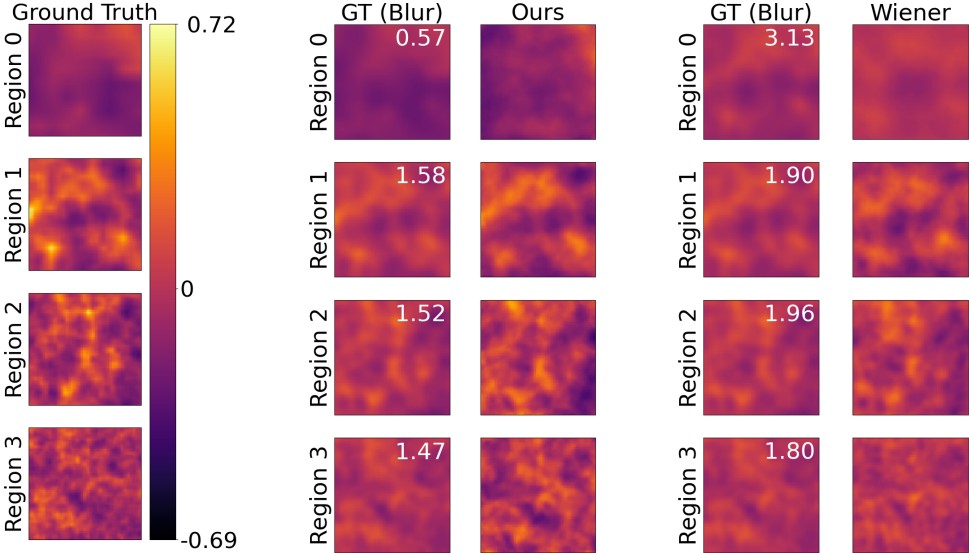

Figure 3: Kinematic weak lensing result: We present 3D reconstruction results of a realistic lensing volume derived from an N-body simulation. Results includes 12 lens planes from redshift $z = 0$ to $z = 1$. To concisely visualize reconstructions, we average every 3 adjacent lens planes to form 4 lensing regions, increasing in distance away from the sensor. Ground truth (GT) lensing regions are shown on the far left. For visual comparison, we blur the ground truth volume with a Gaussian filter in the line-of-sight direction that maximizes the cross-correlation with each reconstructed slice. We then visualize this optimally blurred ground truth volume for each reconstruction, along with the average line-of-sight standard deviation used for each region (top right corner). Our reconstruction corresponds well with a GT volume with significantly lower radial blurring, especially at lower redshift, showing that our method is less susceptible to smearing along the line of sight.

matter in the front half. Thus, we only present the reconstructed volume up to redshift $z = 1$; this is a common practice in 3D mass mapping (Simon et al. (2009), Leonard et al. (2014)).

## 4.2 LARGE-SCALE STRUCTURE RECOVERY WITH KINEMATIC WEAK LENSING

We first use our approach to perform 3D mass reconstruction on realistic kinematic weak lensing measurements of simulated dark matter fields. Results are shown in Fig. 3. We optimize a neural field ensemble as described in Sec. 3.5, taking the median value over each neural reconstruction at sampled points. As a baseline, we compare against the Wiener filter described in Simon et al. (2009). This method has been the most widely used on real data (Simon et al. (2012), Oguri et al. (2018)) and employs a Gaussian prior that requires knowledge of each lens plane's power spectrum.

We present reconstruction results of our overdensity volume on 12 lens planes from redshift $z = 0$ to $z = 1$. To visualize our reconstruction, we average every 3 adjacent lens planes to form 4 lensing regions. Due to large amounts of shape noise, we compare each reconstruction to a blurred version of the ground truth volume. For each reconstructed lens plane we blur the ground truth volume with a Gaussian filter using $\sigma = 2$px in the transverse (angular) directions, and a $\sigma$ in the radial $z$ direction chosen to maximize cross-correlation with each reconstruction. This produces two volumes, optimally blurred to match our reconstruction and the Wiener filtered reconstruction. Additionally, we use the optimal radial blur as a metric for the amount of radial smearing in each reconstruction; a reconstruction with large amounts of smearing along the line of sight would correlate most highly with the ground truth blurred with a wide Gaussian filter in the $z$ direction.

A quantitative analysis of reconstruction performance is shown in Table 1. Downstream applications of 3D mass mapping include the study of mass peaks (areas with high overdensity) and voids (areas with low overdensity); thus, it is important for our reconstruction methods to accurately discriminate between the two. To evaluate the reconstruction quality of both methods, we use the normalized cross-correlation metric without mean subtraction, which harshly penalizes predicting overdensities of the wrong sign. We find that our method outperforms the Wiener filter on both the unblurred and

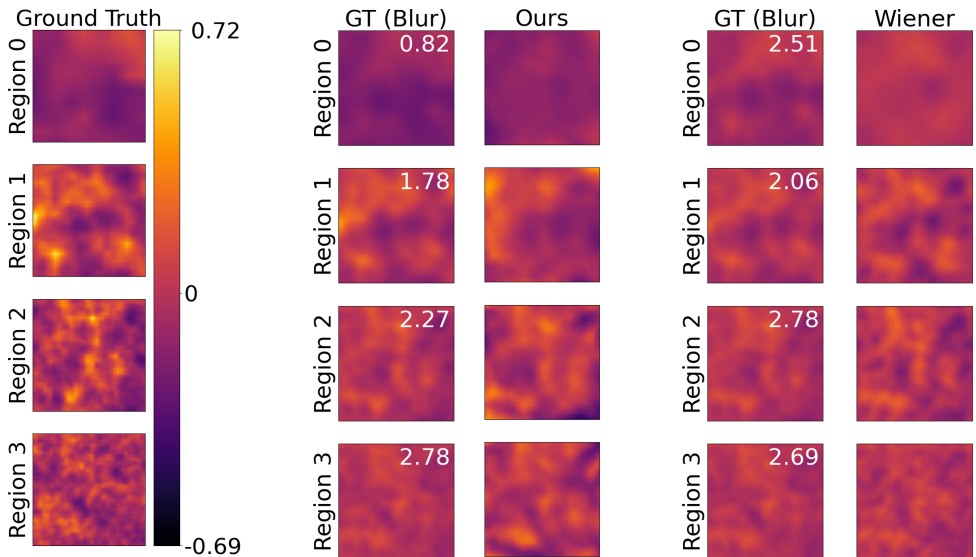

Figure 4: Traditional weak lensing result: We perform 3D reconstruction on a simulated traditional weak lensing survey assuming no information on the intrinsic shapes of galaxies. As in Fig. 3, we use the same ground truth volume and setup, dividing our volume into four redshift regions and generating blurred ground truth volumes for visual comparison. The optimal blur level for each region and method is indicated in the top right corner of each blurred ground truth image. Our reconstruction has less radial blurring than the Wiener filter baseline used in previous weak lensing surveys, and has higher correlation with the ground truth volumes.

optimally blurred ground truth volumes. In addition, we find that the optimally blurred ground truth volume for our reconstruction corresponds to a Gaussian filter with significantly lower width in the radial direction, suggesting that our method better localizes structures along the line of sight; this is especially evident at lower redshifts of the reconstruction.

Table 1: Quantitative Analysis: We report the normalized cross-correlation between the reconstructions and the ground truth, both with and without Gaussian blurring. Cross-correlation values range from $-1$ to $+1$, where higher values signify better reconstruction performance.

| Kinematic WL | CC | CC (Blur) | Traditional WL | CC | CC (Blur) | MNIST Reconstruction | CC | CC (Blur) |
|---|---|---|---|---|---|---|---|---|
| **Ours** | **0.60** | **0.80** | **Ours** | **0.52** | **0.77** | **Ours** | **0.61** | **0.86** |
| Wiener | 0.39 | 0.70 | Wiener | 0.34 | 0.61 | Wiener | 0.52 | 0.68 |

### 4.3 LARGE-SCALE STRUCTURE RECOVERY WITH TRADITIONAL WEAK LENSING

Although kinematic weak lensing measurements show great promise for enabling accurate 3D dark matter reconstructions, kinematic surveys are still in the planning stage (Xu et al. (2023)). Traditional weak lensing surveys do not estimate galaxies' intrinsic shapes, and instead use the fact that they are distributed isotropically and are circular on average. As such, they are subject to much higher levels of shape noise.

In this section we simulate shear measurements with shape noise in the amount expected from a traditional weak lensing survey. Although traditional weak lensing surveys detect a much denser field of galaxies than expected from kinematic surveys, the statistical effect of shape noise only decays with the galaxy density at a rate of $\sqrt{n_{gal}}$ (Schneider et al. (2006)); thus, the combined measurements still contain more noise in aggregate (roughly 2.6x more noise than kinematic). Quantitative results as described in Sec. 4.2 can be found in Table 1. We find that our method again outperforms the Wiener filter baseline and better localizes structures radially, especially at lower redshifts. Thus, we believe that our method could be a powerful tool for analyzing existing weak lensing surveys.

### 4.4 RECONSTRUCTION OF NON-GAUSSIAN STRUCTURES

As our reconstruction in Sec. 4.2 was performed on a low-resolution N-Body simulation, the ground truth volume lacks non-Gaussian features such as sharp edges or peaks, and so was well-suited to

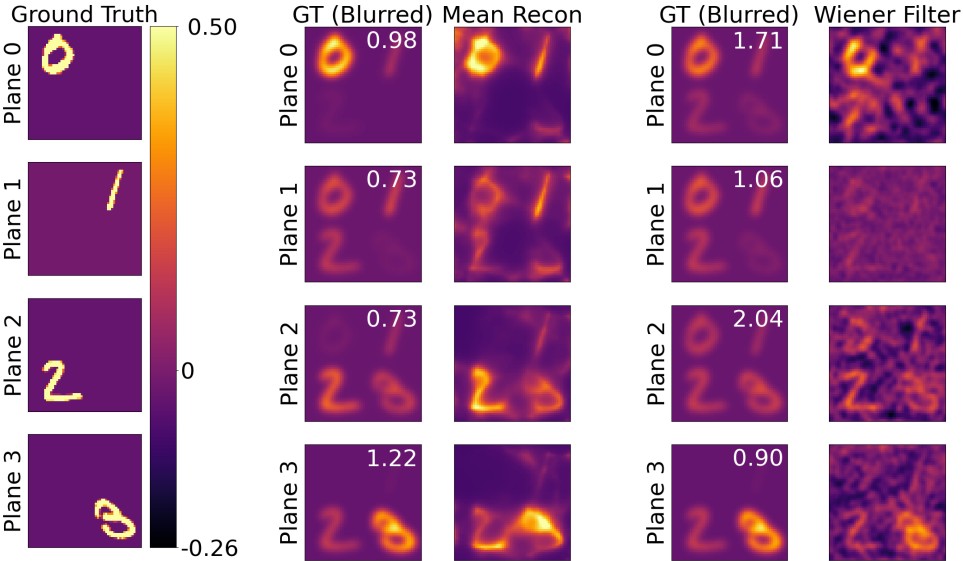

Figure 5: Sensitivity to Non-Gaussianity: In this experiment, we perform reconstruction on a toy lensing field of 4 lens planes, each with an MNIST digit 0 through 3 in one if its corners. Due to the non-Gaussian nature of these images, the Wiener filter which uses a Gaussian prior struggles to reconstruct the lens planes accurately. We blur the ground truth volume to match each reconstructed lensplane as described in Sec. 4.2 and report the optimal $z$ blur (px) in each corner; the Wiener filter exhibits significantly more blurring in low redshifts.

the use of a Gaussian prior. However, recent observations Komatsu et al. (2009) as well as state-of-the-art simulations Pillepich et al. (2018)[2] reveal that some of the most fascinating and scientifically important structures in the universe are highly non-Gaussian. These structures are ill-suited for reconstruction with a Wiener filter, which uses a strong Gaussian prior.

In this experiment we consider a toy example of 4 lens planes, each with an MNIST digit 0 through 3 in a single quadrant, at 4 equally spaced distances from redshift $z = 0$ to 1. We simulate shear measurements with the same galaxy parameters as in Sec. 4.2, where galaxies throughout the volume are lensed only by these 4 lens planes. While the structures of each digit are apparent in both reconstructions, the Wiener filter suffers from heavy amounts of artifacts due to the non-Gaussian nature of the true lens planes. In addition, our method is better able to localize the central locations of each digit in its reconstructions; for the Wiener filter, traces of the digit 2 can be found prominently in the incorrect lens planes. As can be seen in Table 1, our method outperforms the Wiener filter quantitatively on both blurred and unblurred ground truth volumes.

## 5 CONCLUSION

This work presents a method for recovering the 3D cosmic web using a neural field constrained by galaxy shear measurements. First, we demonstrate accurate reconstruction results on simulations of dark matter that outperform the leading baseline method for real data. Second, we show how our method addresses key limitations of the baseline, particularly in capturing non-Gaussian structures within the dark matter field. These structures, which are among the most intriguing and scientifically significant, often exhibit highly non-Gaussian characteristics that are obscured by methods constrained by a Gaussian prior. Furthermore, the continuous and differentiable neural representation underlying our model offers a flexible framework for future advancements, such as incorporating stronger physically motivated constraints, non-discrete galaxy distributions, and more sophisticated spatial priors. Our method offers a flexible yet still powerful solution for 3D dark matter mass mapping, potentially enabling insights into structure formation in the universe and the nature of dark matter itself, two fundamental questions at the heart of physics.

---

[2]Although outputs from these simulations are publicly available, simulation of weak lensing shear measurements from these volumes was prohibitively computationally expensive due to an exponentially greater particle count compared to the simplified N-Body simulations shown in this work.

ACKNOWLEDGMENTS

This work was supported by NSF award 2048237, Google, a Carver Mead New Adventure Fund Award, Stanback innovation Fund Award, Amazon AI4Science Discovery Award, and a Caltech S2I Award. A.L. was supported by the Natural Sciences & Engineering Research Council of Canada (NSERC). We thank Francois Lanusse for his helpful discussions regarding weak lensing and providing us with guidance on the JaxPM code, Carolina Cuesto-Lazaro for her input on dark matter mass mapping applications, and Olivier Doré for helpful discussions about cosmology.

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

# A    APPENDIX

## A.1    N-BODY SIMULATION PARAMETERS

All measurements were simulated in a flat $\Lambda$CDM cosmology with $\Omega_m = 0.3075$, $\Omega_\Lambda = 0.6925$, and $H_0 = 100$ km/s/($h^{-1}$Mpc). We assume that galaxy redshifts are sampled from a distribution with probability density function given by $n(z) = z^\alpha \exp\left(-(z/z_0)^\beta\right)$, where we take $z_0 = 1/1.4$, $\alpha = 2$ and $\beta = 1.5$.

For simulated cosmic shear experiments (Figs 3 , 4), we used a multiple lensplane approximation of the shear, dividing the volume into 18 equally spaced (in comoving distance) lens planes from redshift $z = 0$ to 2. Reconstruction results are shown for the first 12 lensplanes from redshift $z = 0$ to 1. Radial distances to the center of each reconstructed lensplane are given in Table 2, as well as the redshift with respect to the cosmological parameters used in our experiments:

Table 2: Reconstructed Lensplane Redshifts and Distances

| Lensplane | Comoving Distance (Mpc $h^{-1}$) | Redshift $z$ | Present Distance ($10^9$ Light Years) |
|---|---|---|---|
| 1 | 100 | 0.03 | 0.48 |
| 2 | 300 | 0.10 | 1.44 |
| 3 | 500 | 0.17 | 2.41 |
| 4 | 700 | 0.25 | 3.37 |
| 5 | 900 | 0.33 | 4.33 |
| 6 | 1100 | 0.41 | 5.30 |
| 7 | 1300 | 0.49 | 6.26 |
| 8 | 1500 | 0.58 | 7.22 |
| 9 | 1700 | 0.68 | 8.19 |
| 10 | 1900 | 0.78 | 9.15 |
| 11 | 2100 | 0.88 | 10.11 |
| 12 | 2300 | 1.00 | 11.07 |

The MNIST experiment (Fig. 5) was done with 4 equally spaced (in redshift) lensplanes. A table of distances to the center of each of these lensplanes can be found in Table 3.

Table 3: Reconstructed Lensplane Redshifts and Distances

| Lensplane | Comoving Distance (Mpc $h^{-1}$) | Redshift $z$ | Present Distance ($10^9$ Light Years) |
|---|---|---|---|
| 1 | 705 | 0.25 | 3.39 |
| 2 | 1319 | 0.5 | 6.35 |
| 3 | 1848 | 0.75 | 8.90 |
| 4 | 2303 | 1.0 | 11.09 |

## A.2    NEURAL ENSEMBLE SAMPLES

In this section we visualize samples from the Neural Field ensemble used for reconstruction in Sec. 4.2. Due to the ill-posed nature of the reconstruction problem, individual samples can fit the measurement data but vary in structure or contain reconstruction artifacts. However, taking the ensemble mean or median yields a more robust reconstruction that matches well with the ground truth volume.

In addition, we empirically find statistics from the ensemble can be used to perform uncertainty quantification. Results are shown in Fig. 7. Our ensemble can give well-calibrated uncertainty estimates for most reconstruction points, except for outliers at either tail.

## A.3    ADDITIONAL RESULT FOR LARGE-SCALE STRUCTURE RECOVERY WITH KINEMATIC WEAK LENSING

In this section we show a reconstruction experiment with the same simulation parameters as presented in Sec. 4.2 of the main paper, but on a simulated volume initialized with a different random

Table 4: Cross correlation for the kinematic weak lensing experiment presented in Fig. 8

|  | CC | CC (Blur) |
|---|---|---|
| **Ours** | **0.52** | **0.75** |
| Wiener Filter | 0.39 | 0.64 |

Figure 6: Neural ensemble samples: we visualize samples from the neural ensemble used for reconstruction in Sec. 4.2. Individual samples may fit the measurement data equally well but vary in structure or contain artifacts due to the ill-posed nature of the reconstruction. However, taking the ensemble median gives a robust reconstruction which matches well with the ground truth volume.

seed. We show similar findings as in the previous experiment; our method correlates more strongly with both blurred and unblurred ground truth volumes. Moreover, our method exhibits less smearing in the radial direction of reconstructed structures at low redshifts. Reconstruction results are shown in Fig. 8, and quantitative results in Table 4.

## A.4 FULL RECONSTRUCTION VOLUME OF LARGE-SCALE STRUCTURE RECOVERY WITH KINEMATIC WEAK LENSING

In this section we present reconstruction results corresponding to Sec. 4.2 of the main paper. While in Fig. 3 we average lensplanes and optimal z blur levels for sets of 3 adjacent lensplanes for visual brevity, we present the same results in Fig. 9 without averaging.

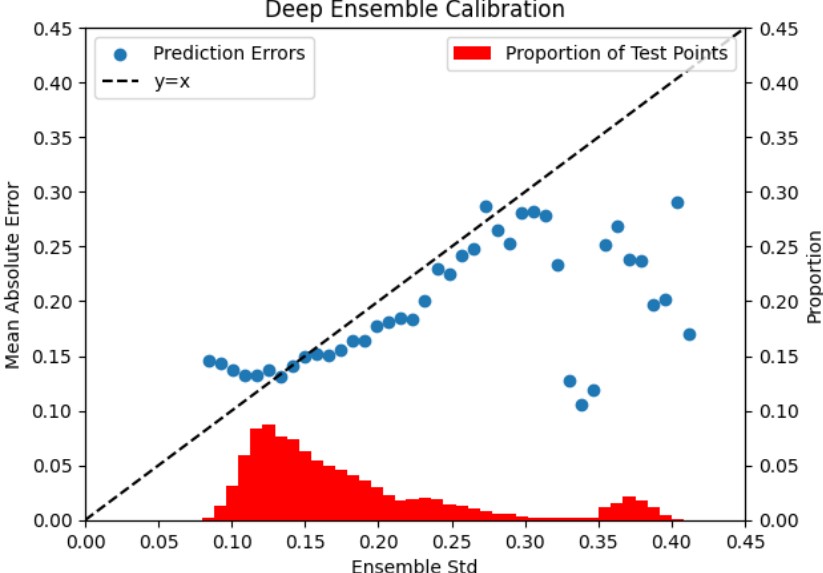

Figure 7: Ensemble Uncertainty Estimates. Although our current ensemble representation doesn't mathematically correspond to a Bayesian posterior, we find that our current method already provides meaningful uncertainty estimates. We present an uncertainty calibration plot for the reconstruction ensemble corresponding to Fig. 3. There is a strong correlation between the ensemble standard deviation and reconstruction error, except for a few outlier regions. In regions with high ensemble variance we find our model is underconfident. In regions with low ensemble variance our uncertainty estimates are slightly overconfident, but the error is still well within 2 standard deviations.

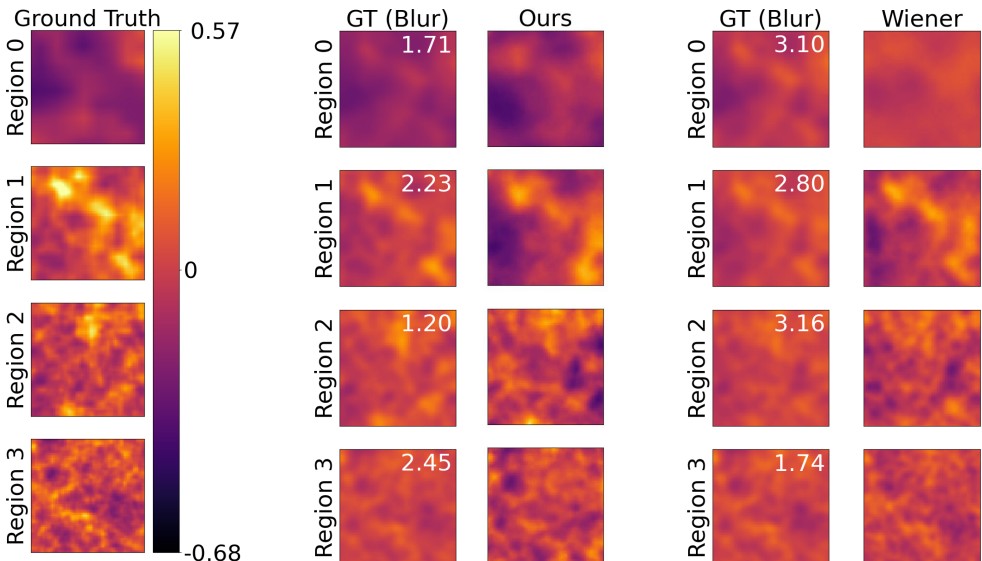

Figure 8: Large-scale structure recovery for an additional simulated kinematic weak lensing survey. We present reconstruction results for a mock kinematic lensing survey with the same parameters as in the main paper, only where the N-Body simulation is initialized with a different random seed. Our method again correlates more strongly with both blurred and unblurred ground truth volumes, and is subject to significantly less radial smearing at low redshifts.

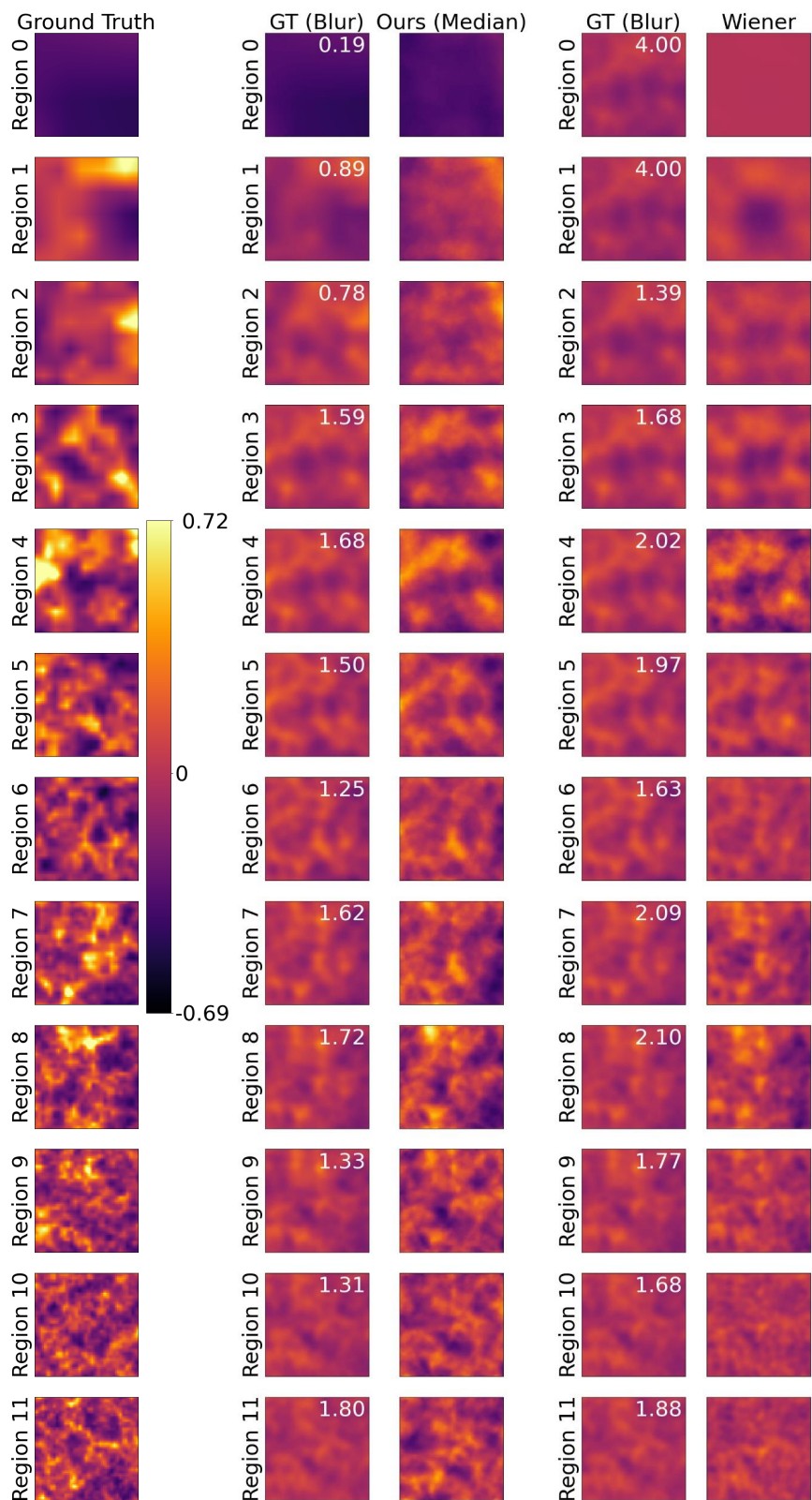

Figure 9: Full Reconstruction volume for Kinematic WL. Reconstruction results in Fig. 3 are shown with sets of 3 adjacent lensplanes averaged. This figure presents the full set of 12 lensplanes without averaging.

