# OpenReview forum: "Revealing the 3D Cosmic Web through Gravitationally Constrained Neural Fields"
_ICLR.cc/2025/Conference — ICLR 2025 Poster_

### Official Review · Reviewer_pR5T · 2024-10-30

**Soundness:** 3
**Presentation:** 3
**Contribution:** 3
**Rating:** 6
**Confidence:** 4

**Summary:**

The paper presents a method for reconstructing the underlying 3D dark matter distribution from weak gravitational lensing observations (specifically, shape distortions of galaxies). This is an important problem in cosmology, and is of contemporary relevance given the number of astronomical surveys which will be measuring galaxy shapes. The overall approach is to model the dark matter distribution implicitly through a neural network (NeRF-like or implicit neural representation), and use a differentiable forward model that models the observed shear field. The authors compare the method to a few traditional approaches for mass mapping, finding it to especially perform well in reconstructing non-Gaussian structures.

**Strengths:**

- **Novel approach and good technical execution:** Using an implicit representation to model the underlying dark matter field along with a differentiable forward model is a very neat but challenging idea, and the authors set it up and executed it very well.
- **Paper straddling AI and fundamental physics:** This is a bit of a meta point, but AI + physics (in particular a sub-field like cosmology) is not particularly mainstream at AI conference so far, in contrast to e.g. AI + bio. This makes presenting a paper like this while being faithful to the domain science particularly challenging, and the authors do a good job in this respect.
- **Simulation realism:** The authors use a realistic set of tools from the cosmology literature, e.g. JaxPM in the differentiable forward model, making it a contribution beyond a simple proof-of-principle.
- **Baseline comparisons:** Comparisons with baselines are well-presented, and highlight the advantage of the method in particular regimes (e.g. in recovering non-Gaussian structures).

**Weaknesses:**

- **Context in current literature and state of the field:** There exists substantial literature in neural approaches to mass mapping; e.g., https://arxiv.org/abs/2201.05561. While the authors compare to traditional approaches like Kaiser-Squires (KS) and Wiener filter, they do not contextualize their work in the more recent ML-based approaches to weak lensing mass mapping. Another example is https://arxiv.org/abs/2206.14820, which uses an implicit neural representation to perform *strong* lensing reconstruction; high-level comparison with existing literature could be improved.
- **Hyperparameter choices:** The authors mention specific hyperparameter choices, e.g. bandwidth of positional encodings L=2 and 5, without further justification. This should be expected to have a significant impact on the results, as it controls the spectral biases of the implicit representation, and should be further expanded upon, possibly including ablations.
- **Role of induced prior:** Traditional approaches typically have a regularization mechanism, often explicit, which allows for mass reconstruction (since the inverse problem is fundamentally ill-posed). In this study, the authors mention that "neural fields have been shown to provide a good implicit prior"; while true in the real-world setting of scene reconstruction, scientific problems are inherently different in nature, and it is not clear a-priori whether the implicit prior induced by the neural field is a good one. The authors approach this empirically, however further understanding of the role of induced priors is necessary for downstream science from the mass maps.

**Questions:**

- The authors use an ensembling approach, taking the median to get a point estimate on the reconstructed mass map. Much of the ongoing literature/effort is focused on _probabilistic_ approaches to mass mapping. The authors mention in a footnote that ensembling can lead to good predictive uncertainties -- can it be expected that this can be used to yield a calibrated distribution over plausible mass maps? If not, are there alternative approaches to this incorporating neural fields, e.g. variational or Bayesian approaches?
- What is the role of bandwidth $L$ and the induced spectral bias in the study? How does the induced prior compare with that assumed in traditional and model ML-based approaches?

---

> ### Author Response · Authors · 2024-11-16
>
> * Thank you for bringing these works applying neural representations to other lensing paradigms; we will be sure to expand our discussion to include these in related works.
> * While we chose the hyperparameter settings empirically, we expect the reconstructions to have a degree of robustness in terms of spectral bias because we are explicitly regularizing the power spectrum of our reconstruction. However, we agree that an ablation on the positional encoding degree would be appropriate and will add one to the appendix.
> * Previous works have been proposed for understanding the implicit prior induced by positional encoding in the context of Neural Tangent Kernel (NTK) theory [1]. Intuitively increasing the bandwidth L should result in an implicit prior with a wider spectrum, allowing the network to fit to higher frequencies. However this is a theoretical result for the infinite-width limit and does not take into consideration our spectral regularization; we are very interested in studying the implicit prior of our method and tuning it appropriately for future work. Empirically, the induced prior of our method appears to be more generalizable than the Gaussian prior of previous approaches.
> * Although our current ensemble representation doesn’t mathematically correspond to a Bayesian posterior, we find that our current method already provides meaningful uncertainty estimates. In our appendix we have added a calibration plot for the experiment shown in Fig. 3. There is a strong correlation between the ensemble standard deviation and reconstruction error, except for a few outlier regions. In regions with high ensemble variance we find our model is underconfident. In regions with low ensemble variance our uncertainty estimates are slightly overconfident, but the error is still well within 2 standard deviations.
>
> [1] https://arxiv.org/abs/2007.05864

---

> ### Comment · Reviewer_pR5T · 2024-11-18
>
> I thank the authors for considering my comments. I still believe that the premise if the study is too empirical for the lensing application -- this is unfortunately a somewhat general concern with application of modern ML methods (which tend to be highly empirical) to the physical sciences (which require a high degree of calibration, control, and robustness, with the current target application being a good example). For example, the role of the implicit induced prior here makes it quite difficult to understand how much of the reconstruction is "real" dark matter, and how much is attributed to the induced prior. While this is not particularly an issue in more traditional application of neural fields to e.g. scene reconstruction, for cosmological applications it is critical. Ablation studies varying over $L$ would be helpful if included, but this is not particularly generalizable -- the effect of bandwidth will be unpredictable when applying the method to a different setting (different survey configuration, survey volume, forward model etc). While shown to be more generalizable than the Gaussian prior of previous approaches, as mentioned in the original review there are other methods that leverage physical priors that are probably more appropriate comparison points. Finally, I appreciate the added study comparing the ensemble error to per-point reconstruction error -- while the ensembling approach seems a bit hacky compared to an in-site probabilistic one, it does give some idea of calibration. Overall, I'm happy to raise my score a notch. Applications to physics problems are challenging, and this is a good effort which I think would be motivating to the conference audience.

---

### Official Review · Reviewer_gGfb · 2024-10-30

**Soundness:** 2
**Presentation:** 2
**Contribution:** 2
**Rating:** 3
**Confidence:** 2

**Summary:**

The article considers the task of reconstructing a 3D map of the universe from weak gravitational lensing obtained in 2D signals. The methods consider first to encode the overdensities using a neural field based on the spherical coordinate and its Fourier decomposition.  The target density field is used for two purposes: (i) to match the power spectrum of the dataset under consideration and (ii) to match the shear observed (after using the transformation from overdensity to shear) in the data. The method is tested on simulated data mimicking the instrumental observations from an N-body simulation. The results show that the method seems better than using Wiener filter for the same task.

**Strengths:**

The authors apply ML techniques to a dataset that is different from the usual benchmarks.

**Weaknesses:**

My overall impression is that we have just an application of ML methods to a cosmological problem, which neither
- make relevant improvement on the side of ML (unless I'm mistaking, it takes out-of-the-box methods)
- make a strong advance in the considered field
I would like the authors to comment on how their work is particularly timely. I'm also thinking that the Wiener filter is not very modern. By quickly looking on the web, we can find more elaborated techniques:
 * sparsity prior: https://www.aanda.org/articles/aa/pdf/2021/05/aa39451-20.pdf, https://arxiv.org/pdf/1801.08945
 * wavelet: https://www.aanda.org/articles/aa/pdf/2006/21/aa2997-05.pdf
And I can imagine there are many others. This work does not provide any comparison to these methods.

The last section that investigate non-Gaussian structure is a bit of a mystery to me. Does the application of the method to MNIST (whatever it means) should be considered of a test of something precise ? In addition,

**Questions:**

- it might be useful to related the various red-shift to a number of years, it is done for z=0 and z=2 but not for z=1.
- The caption of Fig. 3 says: "results includes 12 lens planes from redshift z=0 to z=1", can the authors specify the precise value of the considered z ?
- In addition, the authors say that they average over 3 lens planes. Why ? to which values of z does that correspond ?
- The ground truth images are blurred with a factor that maximize the cross-correlation with the reconstructed picture. It seems a bit weird to do so, it might bias the reader to believe that a reconstruction is better than what it should be.
- The cross-correlation between the reconstructed signal and the true one seems quite small... while we can agree that it is even worse using the Wiener filter, I have the impression that it is working very-well.

---

> ### Author Response · Authors · 2024-11-16
>
> * We will add a table relating redshifts of each lens plane to light years in the appendix. The 12 lens planes are equally spaced in comoving distance, which is standard in works of this type. We average over 3 adjacent lens planes for visual brevity, but we can also add a figure with all 12 planes to the appendix.
> * Because of the nature of noisy single-view backprojection, reconstructions typically suffer from some amount of blurring in the z direction which is not trivial to determine. By finding the optimal blur kernel we are able to estimate this effective blur amount which tells us the resolution at which we can reliably recover structural information.
> * The 3D cross correlation is commonly used in astrophysics for comparing the similarities of 3D fields (see, for example, [2]). It harshly punishes the prediction of overdensities of the wrong sign.
> * We find that a Gaussian assumption can be limiting for 3D dark matter mapping. The purpose of the MNIST experiment was to highlight this point on a toy example for which a Gaussian prior does not match the true field well. Our method performs well in both situations that have Gaussian and non-Gaussian structure. Since non-Gaussian structures, such as galaxy filaments and clusters, are of great scientific interest, we believe our method is better suited for their downstream analysis.
>
> [2] https://academic.oup.com/pasj/article/70/SP1/S26/4097646

---

> > ### Author Response · Authors · 2024-11-23
> >
> > We have now added a table with distance measures for each reconstructed lens plane, as well as a figure visualizing all 12 reconstructed lens planes (without averaging) to the appendix. As the end of the discussion period approaches, we would greatly appreciate it if you could confirm whether our responses have adequately addressed your concerns. If so, we encourage you to consider updating your rating to reflect the improvements made based on your valuable feedback.

---

> > > ### Comment · Reviewer_gGfb · 2024-11-25
> > >
> > > I thank the authors for their careful answer. It is still hard to me to consider this an application of ML that should be published in this venue, I agree this is somewhat subjective and this is confirmed by my confidence score.
> > >
> > > I recognize the effort on the answer, I would put the vote 4, but it is not possible anymore....

---

> > > > ### Author Response · Authors · 2024-11-26
> > > >
> > > > Thank you for the score revision! If you get a chance, would you mind trying again to update your official score in openreview? Reviewers are still supposed to be able to update it at the moment, and we know of other reviewers who updated their scores today.

---

### Official Review · Reviewer_x8ZD · 2024-11-04

**Soundness:** 3
**Presentation:** 4
**Contribution:** 2
**Rating:** 3
**Confidence:** 2

**Summary:**

The paper provides a new method for recovering the 3D cosmic web from galaxy weak lensing signal.
The proposed approach uses a coordinate-based neural field method with positional encoding. (Equation 6).
The reconstruction results are demonstrated in a series of experiments.

**Strengths:**

* The paper proposes a new method for an important physics problem.
* The paper is well-written; I enjoyed reading it.
* The physics background is exceptionally well-presented, and even non-experts can understand the main scientific ideas.
* The proposed method works better than the leading baseline.

**Weaknesses:**

I think this is a great paper, but in my opinion, ICLR might not be the best venue for this.
The new machine learning contributions are quite limited: neural field models and positional encoding have been around for some time, and I didn't find significantly new machine learning contributions in the paper.

Nonetheless, the proposed algorithm works well, and the physics application, estimating the 3D cosmic web, is a very important problem.
Since the most significant contributions of the paper are in astrophysics, I think an astrophysics or cosmology journal would be a better venue for this paper.

**Questions:**

* I would like to know some more details about the number of CPUs/GPUs used for training, memory requirements, and the training and inference time of the proposed method.

* I'm also curious about the next steps. After the 3D cosmic web estimation step is done, what are the next science questions that can be answered with the new results?

---

> ### Author Response · Authors · 2024-11-16
>
> * Training a single neural field model requires 2.3GB of memory and takes 15 minutes for 100k iterations on an A100 GPU. Inference for a single model is almost instantaneous (order of hundredths of a second).
> * 3D dark matter maps are particularly useful for many downstream applications. As highlighted in our introduction, these maps can help to answer questions about the fundamental nature of our universe: Can Dark Matter exhibit trace amounts of gamma ray radiation? How do dark matter structures evolve over time? How can we characterize the primordial distribution of the dark matter field at times close to the Big Bang? Most of these applications specifically target non-Gaussian features of interest, such as galaxy clusters, which makes our method particularly well-suited for downstream science.

---

> ### Author Response · Authors · 2024-11-25
>
> We have written a couple of paragraphs in our global comment which explain why we believe our work constitutes a significant contribution to the machine learning community. As the end of the discussion period approaches, we would greatly appreciate it if you could confirm whether our response has adequately addressed your concerns. If you have any remaining questions, please let us know, and we will do our best to respond within the remaining time.
>
> If your concerns have been addressed, we kindly ask you to consider raising your rating. Thank you again for your time and efforts in reviewing our manuscript.

---

> > ### Comment · Reviewer_x8ZD · 2024-12-02
> > **Re: Official Comment by Authors**
> >
> > Dear Authors,
> >
> > Thanks for your response to my concerns.
> >
> > I still believe that this paper does not have enough new ML contributions. In my opinion, the most important scientific contributions of the paper are in astrophysics and not in machine learning. I do agree, however, that the problem itself can be an important ML application domain.
> >
> > Therefore, I kept my original rating score, but I also lowered my confidence score to indicate that it would be fine with me if the paper gets accepted.

---

> ### Comment · Area_Chair_6q1K · 2024-11-27
> **Response**
>
> Dear Reviewer,
> Do you mind letting the authors know if their rebuttal has addressed your concerns and questions? Thanks!
> -AC

---

### Official Review · Reviewer_p3hC · 2024-11-13

**Soundness:** 3
**Presentation:** 3
**Contribution:** 3
**Rating:** 8
**Confidence:** 3

**Summary:**

***Late review: Apologies my review is in one day late. I will aim to engage as quickly as needed to makeup.***

This very interesting paper considers a very important application, although using some well-established tools in machine learning and computer vision. The paper focuses on the difficult weak-lensing regime, where deflections are too weak to be interpretable reliably for a single galaxy. The idea is that attempting to understand how weak lensing alters (or shears) images of galaxies in dense fields, one could reconstruct a density map of the underlying matter. This is a difficult problem as the matter density is usually analyzed in 2D: the measured shear map is used to recover a projected 2D estimate. The paper aims to solve the inverse problem i.e. to obtain a 3D reconstruction of the dark matter field from 2D images. From a modeling perspective, this becomes challenging as galaxies are observed from a single view. Further, unlensed shapes of known visible sources are themselves not fully understood, introducing a large amount of uncertainty. The main idea of the paper is to use the underlying physics of gravitational lensing to recover a continuous 3D field, with a focus on also capturing non-Gaussian features, which are not easy with traditional methods, which use strong priors. The idea is to use coordinate-based neural fields augmented with the underlying physics (clearly described in figures 1 and 2), with the total loss function described in equation 7. The experiments, over simulated data, are able to show that the method has promise in both reconstructing the 3D matter field and also non-Gaussian features.

**Strengths:**

-- The paper attacks a problem of quite some importance. While the machine learning used is standard, its combination with the underlying physics allows the authors to obtain a promising solution, although it is only tested on simulations (standard for this area of physics at the moment).

-- The paper is well-written, well-structured and easy to follow. While I am not an expert in this specific application, I have worked in some adjacent applications. I think the results support the underlying proposal and motivation well, and are creative in combining standard ML tools with physics-based models.

**Weaknesses:**

-- More of a question: Could the authors elaborate in the paper why are strong Gaussian priors usually made? This will not be obvious to a standard ML audience. I see that it could make sense from the perspective of filtering. But I am not sure I understand why this should drop out from the physics models. I would expect the features to be highly non-Gaussian. I find this confusing whenever I venture to look at papers in this area.

**Questions:**

-- Could the authors elaborate on the training protocols and how difficult it was to get stabilized training? From the writeup, it is not clear how easy it is to get the model to work.

---

> ### Author Response · Authors · 2024-11-16
>
> We train our network through the Adam optimizer with an exponential learning rate decay schedule for a fixed number of iterations. We are able to perform full gradient descent without batching on a single card due to our use of the Fast Fourier Transform in the forward model, so there is no stochasticity after initialization. We have not observed significant instabilities when training our model for experiments. We will make all code and data publicly available for reproducibility.

---

> ### Comment · Area_Chair_6q1K · 2024-11-27
> **Response**
>
> Dear Reviewer,
> While the review was positive and response brief, it would still be helpful to acknowledge and note any other reviews or response influence your score. Thank you.

---

> > ### Comment · Reviewer_p3hC · 2024-11-27
> > **Thanks**
> >
> > Thanks! I did follow the other reviews and responses. I am happy to keep my score as clear accept.

---

### Author Response · Authors · 2024-11-16

Thank you for your helpful comments and questions. We present a “creative” (p3hC) and “novel approach” (pR5T) that attacks “an important physics problem” (x8ZD) with comparisons to baselines that “are well-presented, and highlight the advantage of the method in particular regimes” (pR5T). Reviewer x8ZD mentions that it is a “great paper” that is “exceptionally well-presented” so that “ even non-experts can understand the main scientific ideas.” As highlighted by reviewer pR5T, “presenting a paper like this while being faithful to the domain science [is] particularly challenging, and the authors do a good job in this respect.”

**Machine Learning Contributions (x8ZD, gGfb)**

We firmly believe that our work represents a significant contribution to the ML community. Below, we outline several reasons that we will clarify in the updated paper.

**Neural Fields for Science:** While neural field models with positional encoding have previously been applied to various problems, our work demonstrates how the framework can be incorporated with the underlying physics specific to our problem. We work with noisy measurements from a single viewpoint, and the cosmic web volumes we study have different statistical properties compared to natural scenes. As highlighted by pR5T, this unique context raises important ML questions: How does the implicit regularization in neural fields help estimate solutions to severely ill-posed, underconstrained inverse problems? And how well does positional encoding, effective in natural scenes, generalize to these new scientific contexts? Applications of neural fields was the topic of 2023 ICLR (https://sites.google.com/view/neural-fields) and 2024 ECCV (https://neural-fields-beyond-cams.github.io/) workshops, featuring invited talks that apply neural fields with positional encoding to different 3D reconstruction tasks. We believe that adapting neural fields to a new context with completely different physics represents a significant contribution, and gives value to the ML community.

**Introducing a New Problem to ML:** Additionally, one of the main challenges of interdisciplinary work is effective communication between fields, and we are glad to have expressed our results in a way that is understandable to the ML community (x8ZD, p3hC, pR5T), as this important problem could greatly benefit from their insights and contributions. We also wish to highlight that, to our knowledge, this is the first time that a comparison of 3D mass mapping methods has been done. We will make all code and datasets available so that this problem can be further studied by the ML community.

**A Stepping Stone to Future Work:** We believe our work is a necessary stepping stone to future work that would take advantage of the neural representation we’ve presented. One direction we are actively pursuing is leveraging the neural representation to obtain efficient uncertainty quantification, which, as pR5T notes, is important for downstream science. While probabilistic approaches to 2D mass mapping are an active field of study, to our knowledge none have yet been proposed for 3D mass mapping, possibly due to computational intractability arising from the extra dimension. Even in the simple case of the Wiener filter, which has an analytic posterior, computing this posterior with a full covariance becomes intractable in 3D. In future work, we plan to build on the theory from [1], which demonstrates how neural representations can play an essential role in enabling efficient posterior estimation. The method we present serves as a solid foundation for further progress in this important direction.

**Wiener Filter Baseline (p3hC, gGfb)**

As gGfb highlights through various references, there have been recent developments in the field of 2D mass mapping. However, these methods are not directly applicable to 3D. Currently, there are no publicly available 3D mass mapping codes in working condition. While one code exists, it is non-functional, and the authors were unable to provide a working version. To the best of the authors' knowledge, the Wiener filter remains the only 3D method that has been applied to real cosmic shear data, with its most recent usage documented in 2018 [2]. Although code for this method was also not publicly available, we implemented our own transverse Wiener filter to establish a baseline.

In regards to p3hC’s question about Gaussian priors: On scales much larger than those we are targeting, the overdensity field of the universe can be described as Gaussian [3]. This coupled with the simplicity of the Gaussian prior makes it a popular choice for reconstructions of this type. However, non-Gaussianity present on small cosmic scales motivates our more general ML-based approach.

[1] https://arxiv.org/abs/2007.05864

[2] https://academic.oup.com/pasj/article/70/SP1/S26/4097646

[3] https://ned.ipac.caltech.edu/level5/March01/Coles/Coles4.html

---

### Meta-Review · Area_Chair_6q1K · 2024-12-19

**Metareview:**

This paper studies the problem of reconstructing the 3D dark matter field from 2D observations. The method uses a coordinate-based neural field with positional encodings to represent the matter field which is then passed through a physics-based differentiable forward model to obtain a shear field which can be compared to observations.

Strengths: This paper addresses an important problem in astronomy. The technique is a novel one in weak lensing, although the underlying neural field methods are not new, and is an interesting combination of ML methods with a physics-based simulator.  The evaluations are promising and show good results against reasonable baselines on simulated data. The paper is well-written and contains a good background on the application area for a general audience.

Weaknesses: The main the noted weakness of the paper is limited innovation the ML methodology which uses standard techniques, although they have not been applied to this physics problem before.  Another noted limitation of the method is limited ability to determine the impact of implicit priors in the reconstruction.

Conclusion:    Despite the range of scores, reviewers broadly agreed on the merits and weaknesses of the paper.  All reviewers actually agree that the paper is of high quality, but question whether ICLR is the right venue for publication since the paper does not make any new contributions to core ML methodology and is largely an application of ML to a new domain.  There is not a clear cut consensus here, and I appreciate the reviewers openness and flexibility during the discussion phase. In the end, I tend to believe that scientific applications papers are important to the field and figuring out how to deploy even existing methods in new domains is worthwhile to our community.

**Additional Comments On Reviewer Discussion:**

Reviews for the paper were quite split with two reviewers on the side of rejecting (x8ZD,gGfb) and two on the side of acceptance (pR5T, p3hC).  Despite the range of scores, reviewers broadly agreed on the merits and weaknesses of the paper.  From the discussion phase during which all reviewers contributed, all reviewers actually agree that the paper is of high quality, but question whether ICLR is the right venue for publication since the paper does not make any new contributions to core ML methodology and is largely an application of ML to a new domain. Both x8ZD and gGfb indicated that despite their negative score, they are not dead set against the paper being accepted.  pR5T who raised their score from 5 to 6, pointed out the contributions to astronomy probably do not yet meet the bar for publication in that field. I cannot evaluate that point.  While there is not a clear cut consensus here, I appreciate the reviewers openness and flexibility. In the end, I tend to believe that scientific applications papers are important to the field and figuring out how to deploy even existing methods in new domains is worthwhile to our community.

---

### Decision · Program_Chairs · 2025-01-22

Accept (Poster)